# Mono(imidazolin-2-iminato) Hafnium Complexes: Synthesis and Application in the Ring-Opening Polymerization of ε-Caprolactone and *rac*-Lactide

**Maxim Khononov [1], Heng Liu [2,*], Natalia Fridman [1], Matthias Tamm [3] and Moris S. Eisen [1,*]**

[1] Schulich Faculty of Chemistry, Technion-Israel Institute of Technology, Technion City 3200003, Israel

[2] Key Laboratory of Rubber-Plastics, Ministry of Education/Shandong Provincial Key Laboratory of Rubber-Plastics, Qingdao University of Science & Technology, Qingdao 266042, China

[3] Institut für Anorganische und Analytische Chemie, Technische Universität Braunschweig, Hagenring 30, 38106 Braunschweig, Germany

[*] Correspondence: dr_hengliu@163.com (H.L.); chmoris@technion.ac.il (M.S.E.); Tel.: +972-4-8292680 (M.S.E.)

**Abstract:** Mono-substituted imidazolin$^X$-2-iminato hafnium(IV) complexes (X = $^i$Pr, $^t$Bu, Mesityl, Dipp) were synthesized and fully characterized, including solid-state X-ray diffraction analysis. When the X group is small ($^i$Pr), a dimeric structure is obtained. In all the monomeric complexes, the Hf-N bond can be regarded as a double bond with similar electronic properties. The main difference among the monomeric complexes is the cone angle of the ligand, which induces varying steric hindrances around the metal center. When the monomeric complex of mono(bis(diisopropylphenyl)imidazolin-2-iminato) hafnium tribenzyl was reacted with three equivalents (equiv) of $^i$PrOH, the benzyl groups were easily replaced, forming the corresponding tri-isopropoxide complex. However, when BnOH was used, dimeric complexes were obtained. When five equivalents of the corresponding alcohols (BnOH, $^i$PrOH) were reacted with the monomeric complex, different dimeric complexes were obtained. Regardless of the high oxophilicity of the hafnium complexes, all complexes were active catalysts for the ring-opening polymerization (ROP) of ε-caprolactone. Dimeric complexes **5** and **6** were found to be the most active catalysts, enabling polymerization to occur in a living, immortal fashion, as well as the copolymerization of ε-caprolactone with *rac*-lactide, producing block copolymer PCL-*b*-LAC. The introduction of imidazolin-2-iminato ligands enables the tailoring of the oxophilicity of the complexes, allowing them to be active in catalytic processes with oxygen-containing substrates.

**Keywords:** hafnium complexes; imidazolin-2-iminato; polycaprolactone; polylactide

## 1. Introduction

Currently, polymers play an important role in our everyday lives, making them convenient and leisurely in many ways. Polymers are employed as building blocks for a wide variety of industrial and domestic applications. Their outstanding performances are enabled by their diverse properties, which can be altered and tailored depending on the polymerized monomer [1]. However, one of their disadvantages is that once a plastic product is out of usage, it accumulates as waste. Most plastics are made from non-biodegradable materials, and the process of decomposition lasts for decades [2,3]. In addition to the challenge of selectively obtaining a polymer with a defined stereochemistry, the consumption of polymer-based materials and the rate of plastic waste have increased considerably within the last two decades, resulting in major environmental issues and increasing the need for research on biodegradable and ecologically friendly materials [2,4–6]. In order to achieve the preparation of biodegradable materials, it is necessary to develop novel catalytic systems that allow for the synthesis of these polymers with high yields and without additional byproducts.

To ensure their use, biodegradable polymers need to have similar or even superior properties to those of traditional non-biodegradable polymers, such as analogous mechanical properties combined with a complete degradation process in the presence of micro-organisms [7–9]. Biodegradable polymers can be natural polymers or synthetic polymers. Starch, chitin, and cellulose are examples of natural polymers. On the other hand, synthetic biodegradable polymers include poly($\varepsilon$-caprolactone) (PCL), polylactides (PLA), etc. PCL and PLA are biocompatible polymers used for medical and pharmaceutical applications [10–12]. These aliphatic polyesters are formed by the convenient method of ring-opening polymerization (ROP) of their monomeric cyclic esters [13]. Studies have demonstrated the impact of the obtained molecular weight ($M_n$) of the polymer on its degradation rate and mechanical strength [12]. ROP of these aliphatic esters can be performed using metal complexes as catalysts based on main group metals, transition metals, lanthanides, and actinides [14–67].

Despite the wide variety of complexes used today for ROP, complexes containing hafnium(IV) have not yet been fully developed as compared to other metal complexes, such as aluminum and zinc complexes. Hafnium complexes are often used in the field of homogeneous catalysis, such as for the hydroboration of aldehydes, ketones, and carbodiimides [68]; in the catalytic addition of alcohols to carbodiimides; and the polymerization of $\alpha$-olefins [69–84]. However, in the ring-opening polymerization of cyclic esters, the variety of hafnium metal complexes applied as catalysts is limited [20,85–91]. This limitation can be associated with the formation of a catalytically inactive, stable hafnium–oxygen bond (Hf-O; 801 kJ/mol) [92]. Thus, the significance of the selection of the metal and the anchoring ligand is substantial, as they directly affect the activity and the obtained polymer structure [89–91]. To deal with the oxophilicity of hafnium, a super-basic imidazolin-2-iminato ligand was introduced to reduce its electrophilic nature. Imidazolin-2-imides (Structure A, Scheme 1) represent highly nucleophilic ligands that can act as a $2\sigma$, $4\pi$ electron donors (Structure B, Scheme 1) toward early transition metals in high-oxidation states [93,94]. Because the ligand exhibits strong donating abilities and is an isolobal, monodentate analog to cyclopentadienyl (Cp), it is expected to increase the electron density at the hafnium atom, resulting in a more active catalyst toward oxygen-containing substrates, such as the cyclic esters $\varepsilon$-caprolactone and lactide. In addition, we expect that the formation of the imidazolin-2-iminato hafnium bond will result in short Hf-N bond distances and large, almost linear Hf-N-C angles, suggesting a higher M-N bond order [68,95–102].

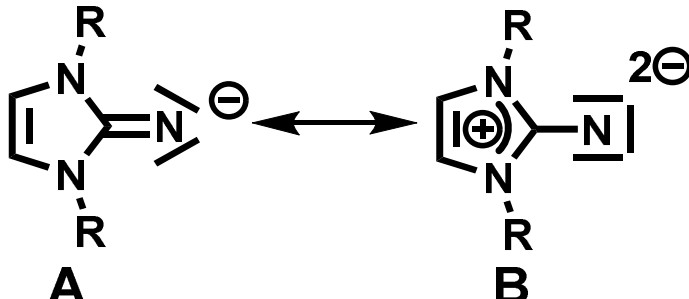

**Scheme 1.** Resonance structures of the imidazolin-2-iminato ligand.

In this work, we focused on the synthesis and characterization of various hafnium(IV) complexes containing the imidazolin-2-iminato ligand. These complexes were found to be catalytically active in a saturated environment of oxygen-containing substrates (cyclic esters $\varepsilon$-caprolactone and/or lactide), enabling the synthesis of biodegradable polymers and copolymers with high yields and without further byproducts.

## 2. Results and Discussion

### 2.1. Synthesis and Characterization of Hafnium Complexes

Mono(imidazolin-2-iminato) hafnium(IV) complexes **1**–**4** (Scheme 2) were obtained according to our previously published procedure for the preparation of complexes **2** and **4** [68]. The reaction of a toluene solution of the neutral imidazolin-2-imine was added to a toluene solution of 1 equiv of the homoleptic hafnium tetrabenzyl complex (HfBn$_4$). The reaction mixture was stirred at room temperature for 16 h, followed by solvent removal under a vacuum. Each complex was crystallized at −35 °C from a mixture of toluene and hexane (10:1). Complex **1**, (Im$^{i\mathrm{Pr}}$N)HfBn$_3$, (Figure 1) crystallizes in the monoclinic space group C2/c as a dimeric complex with two hafnium centers. Each metal atom is surrounded by three benzyl groups and two bridging imidazolin-2-iminato ligands bound through the exocyclic-N1 moieties. The Hf-N bond lengths are 2.118(5) Å for Hf1-N$_1$ and 2.175(5) Å for Hf1$^{\#1}$-N1, and the Hf-N-C bond angles are 129.5 (4)° and 127.6 (4)° for Hf1 and Hf1$^{\#}$1, respectively. We expected to obtain a short Hf-N bond length and a linear-angle Hf-N-C, as previously observed in actinide, zirconium, and titanium complexes using this family of ligands [26,96,99–114]. Hence, we suspected that the use of the small isopropyl moieties enabled the formation of the dimeric structure and distribution of the electron density of the ligands over two metal centers. Accordingly, by tailoring the bulkiness of the substituents on the imidazoline skeleton (Im$^x$N) instead of Im$^{i\mathrm{Pr}}$, we decided to use the larger Im$^{t\mathrm{Bu}}$, Im$^{\mathrm{Mes}}$, and Im$^{\mathrm{Dipp}}$ moieties to prevent the formation of the dimer structure (Scheme 2). Complex **3** crystallizes in a monomeric structure with a distorted tetrahedral geometry around the metal atom (Table 1). Complex **3** (Figure 2), as previously reported for complexes **2** and **4** [68], crystallized in the triclinic space group P-1. Complexes **2**–**4** exhibit short Hf-N bond lengths of 1.925(4) Å, 1.909(5) Å, and 1.911(5) Å, respectively, as compared to other hafnium complexes bearing amido ligands [115–118]. In addition, the similarity among the exhibited Hf-N bond lengths on complexes **2**–**4** indicates a minimal influence of the substituents on the electronic properties in these complexes. The selected bond lengths (Å) and angles (°) for complexes **1**–**4** are summarized in Table 1.

**Scheme 2.** Synthesis of mono(imidazolin-2-iminato) hafnium(IV) complexes **1**–**4**.

As expected, complexes **2**–**4** display shorter Hf-N bond lengths as compared to complex **1** by ~0.2 Å, given the difference between single- and double-bond motifs. The long Hf-N bond length in complex **1** is attributed to the four-electron donation of exocyclic nitrogen to two different metal atoms (Hf1-N1-Hf1$^{\#}$) instead of one metal center. This dimeric structure is the result of using less crowded ligands for the complexation of the highly electrophilic hafnium center.

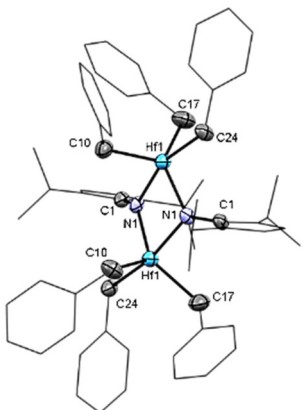

**Figure 1.** ORTEP drawing of complex **1** [(Im$^{i\text{Pr}}$N)$_2$Hf$_2$(Bn)$_6$]. Hydrogen atoms are omitted for clarity.

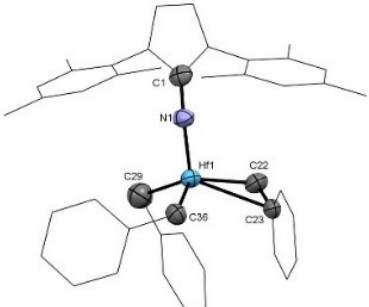

**Figure 2.** ORTEP drawing of complex **3** [(Im$^{\text{Mes}}$N)Hf(Bn)$_3$]. Hydrogen atoms are omitted for clarity.

Comparing complex **3** with complexes **2** or **4**, complex **2** displayed a larger, almost linear Hf-N$_{\text{exocyclic}}$-C$_{\text{ipso}}$ angle, with values of 176.3(3)°, 164.1(4)°, and 172.2(3)° for complexes **2**, **3**, and **4**, respectively. The disposition of the benzyl rings in complex **3** enables an agostic interaction with the *ipso* carbon of one of the benzylic rings (Hf-CH$_2$-C$_{\text{aromatic}}$ = 88.9(4)°). Despite the similarities among complexes **2**–**4**, a noteworthy difference is the cone angle—124°, 155°, and 262° for complexes **2**–**4**, respectively—as expected given the variable bulkiness of the imidazolin-2-iminato ligands in the corresponding complexes. Furthermore, an outstanding difference between symmetric and asymmetric complexes can be observed by comparing complex **3** with a similar asymmetric complex containing only one mesityl and one methyl group instead of one mesityl substituent at each nitrogen atom (Figure 3). The bulkiness of the abovementioned ligand induces a longer Hf-N bond length in complex **3** of ~0.1 Å. In addition, this asymmetric complex, as expected, displays a small cone angle (125°), enabling the ligand to get closer to the hafnium center [69].

**Figure 3.** Schematic drawing of the asymmetric complex, imidazolin$^{\text{Mes,Me}}$-2-iminato hafnium.

A close examination of the exocyclic iminato, $N_{exocyclic}$-$C_{ipso}$ bonds in complexes **1–4** indicates similar bond lengths of 1.330(8), 1.323(6), 1.297(6), and 1.299(5) Å. These similarities suggest a minimal electronic effect of the ligands on the metal center. Comparison of the N $N_{exocyclic}$-$C_{ipso}$ bond lengths indicates the relative electron donation from the ring to the metal center among the complexes (the neutral ligand Im$^{Dipp}$ has an N $N_{exocyclic}$-$C_{ipso}$ bond length of 1.279(3)Å).

Hf(IV) complexes **1–4** showed a high resemblance to the analogous Ti(IV) and actinide (U, Th) complexes compared to the corresponding $N_{exocyclic}$-$C_{ipso}$ bond lengths and M-$N_{exocyclic}$-$C_{ipso}$ angles [94,105]. However, in comparison with analogous lanthanide (Lu, Gd, Yb, and Sm) complexes, the Hf(IV) complexes have longer $N_{exocyclic}$-$C_{ipso}$ bond lengths, suggesting a higher electron donation from the ligand to the metal, probably due to the difference in electrophilicity between the mentioned groups [119].

Mono(imidazolin-2-iminato) hafnium(IV) complexes **5** and **6** were obtained by reacting a toluene solution of 1 equiv Im$^{Dipp}$NHfBn$_3$ (**4**) and 3 equiv of the alcohol ROH (R = benzyl (Bn) or $^i$Pr). The reaction mixture was stirred at room temperature for 24 h, and the solvent was removed under a vacuum. Complex **5** (Figure 4) crystallized at −35 °C from a mixture of toluene and hexane (1:10).

**Table 1.** Selected bond lengths (Å) and angles (°) for **1–4**.

| Bond Length (Å) and Angle (°) | 1 | 2 | 3 | 4 |
|---|---|---|---|---|
| Hf-N1 | 2.118(5) | 1.925(4) | 1.907(4) | 1.911(4) |
| Hf-N1$^{#1}$ | 2.175(5) | | | |
| Hf-$C_{Benz}$ | 2.268(7) | 2.267(5) | 2.229(6) | 2.254(5) |
| N-$C_{ipso}$ | 1.330(8) | 1.323(6) | 1.297(6) | 1.299(5) |
| Hf-$C_{Benz}$ | 2.262(8) | 2.270(5) | 2.251(6) | 2.279(5) |
| Hf-$C_{Benz}$ | 2.326(5) | 2.290(6) | 2.278(5) | 2.281(4) |
| Cone angle | 128 | 124 | 155 | 262 |
| N1-Hf1-N1$^{#1}$ | 76.26(19) | | | |
| Hf1-N1-Hf1$^{#1}$ | 102.15(19) | | | |
| C1-N1-Hf | 129.5(4) | 176.3(3) | 164.1(4) | 172.2(3) |
| $C_{Benz}$-Hf-$C_{Benz}$ | 91.5(3) | 104.2(2) | 107.1(2) | 102.7(2) |
| $C_{Benz}$-Hf-$C_{Benz}$ | 98.5(3) | 113.3(19) | 109.6(2) | 117.4(2) |
| $C_{Benz}$-Hf-$C_{Benz}$ | 118.2(3) | 121.48(19) | 122.3(2) | 121.3(18) |

In contrast to complex **4** [112], complex **5** crystallized in a dimeric form in the monoclinic space group $P2_1/c$. In complex **5**, each hafnium center is coordinated to one Im$^{Dipp}$N ligand, two alkoxy moieties, and two bridging alkoxy motifs. The exocyclic N atom of the imidazolin-2-iminato moiety exhibited a slightly longer bond length with the hafnium center of 1.971(7) Å as compared to 1.911(5) Å in complex **4**. This elongation is the result of the increased electron density at the metal center caused by the oxygen ligands, decreasing the need for π donation from the nitrogen donor ligand.

The $N_{exocyclic}$-$C_{ipso}$ bond length is slightly shorter in complex **5** (1.251(10) Å) than in complex **4** (1.300(6) Å), indicating almost no electronic involvement of the ring in stabilizing the metal center. Both complexes display almost linear angles with respect to the Hf-N-$C_{ipso}$ vector. The strength of the Hf-N bond is not cleaved by protonation, even in the presence of alcohol.

In complex **5**, the two Im$^{Dipp}$ moieties are almost parallel to one another (Figure 5) and disposed with an angle of 111° between the ligand and a vector between the two metal centers ($N_{exocyclic}$-Hf$_1$-Hf$_1$), displaying an antiperiplanar, chair-like dihedral structure (L-M-M-L). The aromatic rings in each of the imidazolin-2-iminato ligands are almost coplanar; therefore, two of the isopropyl groups in each ligand are directed toward the metal centers (Figure 5). This disposition forces the other benzyloxy moieties to point "outside" the core of the dimer, maximizing the distance from other ligands and reducing the steric repulsion to a minimum.

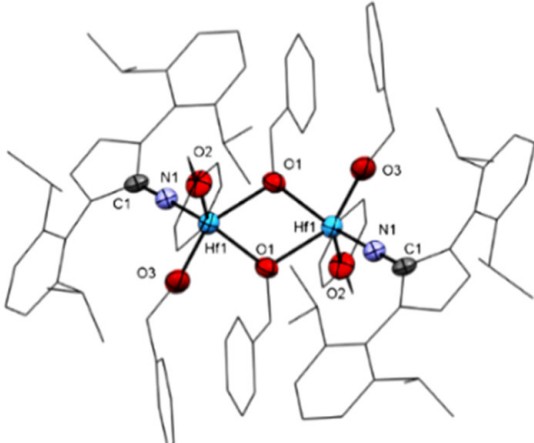

**Figure 4.** ORTEP drawing of **5** [(Im$^{DiPP}$N)$_2$Hf$_2$(OBn)$_6$]. Hydrogen atoms are omitted for clarity. Selected bond lengths (Å) and angles (°): Hf-N1 = 1.971(7), Hf-O1 = 2.136 (5), Hf1-O1 = 2.198(6); Hf-O2 = 1.936(6), Hf-O3 = 1.958(7); N1-C1 = 1.251(10) Å; Hf-N1-C1 = 176.3(6), Hf-O1-Hf$^1$ = 113.2(2); O1-Hf-O1$^1$ = 66.8(2) (°).

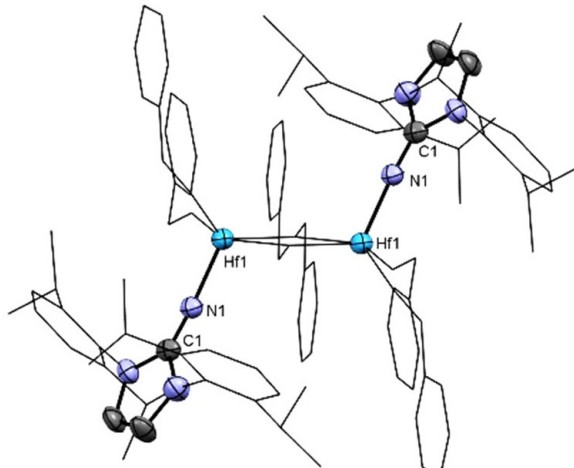

**Figure 5.** ORTEP drawing of **5** [(Im$^{DiPP}$N)$_2$Hf$_2$(OBn)$_6$] showing the chair-like structure. Benzyloxy and imidazolin substituents atoms are shown in wireframe mode to clarify the structure. Hydrogen atoms are omitted for clarity.

Complexes **7** (Figure 6) and **8** (Figure 7) were obtained via a similar procedure with the addition of 5 equiv of the respective alcohols ROH instead of 3 equiv as mentioned in Scheme 3. When an excess of ROH was utilized, the imidazolin-2-iminato ligand was detached from the hafnium complex, as shown in complexes **7** and **8**. Crystallographic data for complexes **5**, **7**, and **8** are summarized in the Supplementary Materials.

Complex **7** crystallizes in the space group $P2_1/c$. The complex is dimeric, and each Hf metal has a distorted octahedral coordination structure with three terminal and three bridging alkoxy moieties. The Hf-O$_{bridge}$ distances are between 2.166 and 2.208 Å and slightly larger than the terminal alkoxo moieties (1.940(3)–1.984(3) Å). Because the formula of the complex is [Hf$_2$(OBn)$_8$(BnOH)], the X-ray data indicate that the proton is scrambled among the alkoxy moieties. The angles of the bridging Hf-O$_{bridge}$-C$_{benzyl}$ (ranging 121.89°–132.39°) than the terminal equivalent angles (ranging 138.17°–170.64°), as expected.

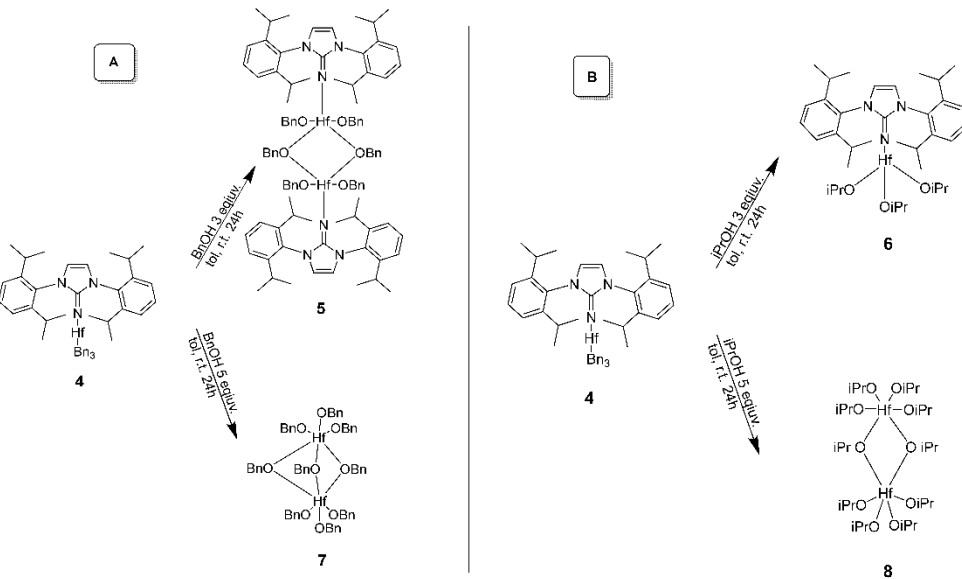

**Scheme 3.** Synthesis of hafnium(IV) complexes **5, 7** (**A**), **6**, and **8** (**B**). In complexes **7** and **8**, one alkoxo moiety is protonated, and the hydrogen is not allocated, as it is very dynamic, as observed by NMR spectroscopy.

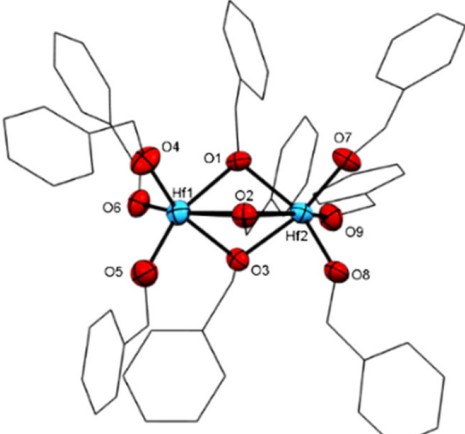

**Figure 6.** ORTEP drawing of **7** [Hf$_2$(OBn)$_8$(BnOH)]. Hydrogen atoms are omitted for clarity.

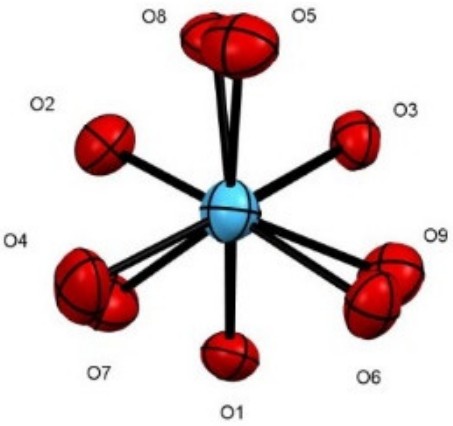

**Figure 7.** ORTEP drawing of **7** [Hf$_2$(OBn)$_8$(BnOH)] from the Hf-Hf vector view showing only the oxygen atoms.

Comparison of the conformation of the benzyloxy moieties bound directly to the metal center, Hf-O-C-C$_{phenyl}$, in complexes **5** and **7** shows that they differ in terms of the values of the dihedral angles (Figure 5). The bridging alkoxy moieties are disposed in an *anti* direction to the imidazoline motif. In complex **7**, an almost perfect *anti*-conformation is observed for the oxygen moieties; O$_1$, O$_2$, and O$_3$ are *anti* to O$_4$, O$_5$, and O$_6$, respectively (Figure 8).

Complexes **5** and **7** both have bridging BnO motifs. A close inspection of the bond length between the C$_{Benz}$-O moieties indicates that in complex **5**, this bond is slightly longer than in complex **7** by 0.078 Å, presumably due to the difference in electron donation of the oxygen moieties as compared to the imidazolin-2-iminato ligand to the metal centers.

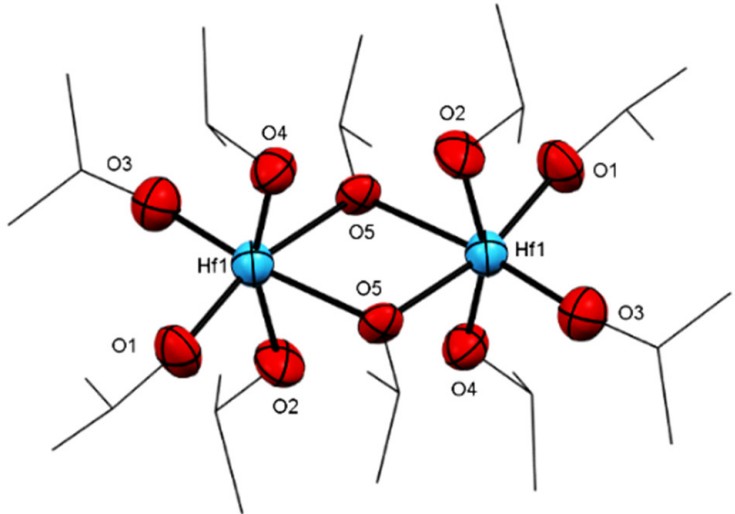

**Figure 8.** ORTEP drawing of **8** [Hf$_2$(O$^i$Pr)$_8$·($^i$PrOH)$_2$]. Isopropyl atoms are shown in wireframe mode to clarify the structure. Hydrogen atoms are omitted for clarity.

In complex **5**, the hafnium atoms are bridged by only two alkoxy units, whereas in complex **7,** the hafnium atoms are bridged by three alkoxy units. This coordination difference is the result of the bulkiness and the steric repulsion of the alkoxy moieties with the large imidazolin-2-iminato ligand. In the latter complex, due to the high oxophilicity of the metal center, an additional neutral molecule of alcohol is also coordinated.

Furthermore, by using a similar procedure to prepare complex **7** but with less alcohol, $^i$PrOH, the dimeric structure of complex **8** is obtained (Figure 8). In complex **8**, four isopropoxide units are connected directly to each hafnium, and only two isopropoxides act as bridging units. The complex obtained has the formula [Hf$_2$(O$^i$Pr)$_8$·($^i$PrOH)$_2$] indicates that fewer steric isopropoxy moieties enable the coordination of additional neutral alcohol molecules. In complex **8**, the oxygens (O1, O3, and O5) and both hafnium are disposed in the same plane, whereas the O2 and O4 are almost perpendicular to that plane, inclined toward the inner "dimer core", with an angle of 86.6(4)° for O4-Hf1-O5 and of 78.0(4)° for O2-Hf1-O5 (Figure 9). The O2-Hf1 bond length is the longest among other $^i$PrO-Hf in the same complex.

In addition, in complex **8**, the four axial alkoxy ligands display smaller bending angles (Hf-O2-C4 = 127.93, Hf-O4-C10 = 129.29 Hf-O2-C4= 127.93 Å) than the four alkoxo moieties on the plane of the molecule (Hf-O1-C2 = 170.61, Hf-O3-C7 = 170.77 Å). Therefore, we suggest that the two protons are involved in a hydrogen bond between the axial alkoxy moieties O2-H-O4 (O2-O4 = 2.760 Å as compared to O1-O3 = 2.918 Å), and the lack of bending in the plane of the molecule seems to be a result of steric interactions.

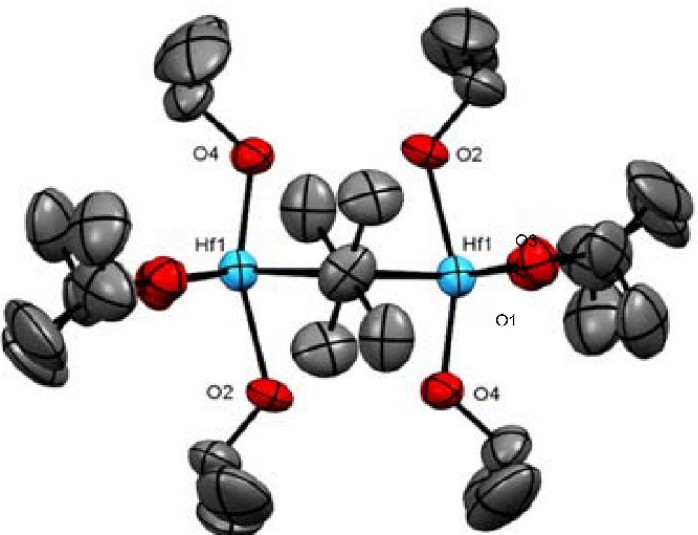

**Figure 9.** ORTEP drawing of **8** [Hf$_2$(O$^i$Pr)$_8$·($^i$PrOH)$_2$]. Hydrogen atoms are omitted for clarity.

*2.2. Ring-Opening Polymerization Studies*

The ring-opening polymerization of a cyclic lactone or lactide monomer is a desired, straightforward method to afford biodegradable polyesters due to the absence of any by-products. We were interested in investigating the reactivity of the oxophilic hafnium(IV) complexes **1–4** as catalysts toward oxygen-containing monomers. Complexes **1–4** exhibit similar electronic properties based on their similar bond lengths; however, their steric properties differ considerably, i.e., a smaller cone angle of the Im$^{tBu}$N ligand in complex **2**, followed by an increase in the cone angle of the Im$^{Mes}$N ligand in complex **3** and a very large cone angle of the ligand Im$^{Dipp}$N in complex **4**. We predicted that a complex with a smaller cone angle would have a more exposed metal center. Hence, a deactivation of the complex will be inevitable as compared to a complex with a larger cone angle and a more well-shielded complex, as in complex **4**. The deactivation is expected to occur due to oversaturated coordination by the incoming monomers and/or via the coordination of the growing polymer chain(s) that will rotate toward the active site and block other coordination sites. In a complex with a large cone angle, it is expected that monomer insertion/polymer growth will be the preferred operative process. To corroborate our expectations, we studied such effects, and these complexes were utilized in the ROP of *ε*-caprolactone. Polymerization reactions were performed at room temperature and with varying complex-to-monomer ratios: 1:100, 1:1000, and 1:2000. The polymerization results are summarized in Table 2. As expected, the activity of complexes **1** and **2** are similar. With increased monomer content, the activity increases, obtaining the best TOF at a ratio of 1:1000. Increased monomer contents do not induce increased activities but transesterifications for complexes **1–3**. For example, using complexes **3** at a ratio of 1: 100, yields a polymer with $M_n$ of 3450 and a polydispersity (PDI) of 2.4. At a ratio of 1:1000, the polymer $M_n$ increased to 3700, whereas the polydispersity increased to 4.3. At a ratio of 1:2000, the polymer $M_n$ was 3950, and the polydispersity was increased to 6.4. In contrast, the polymerization of caprolactone using the oxophilic complex (Im$^{Dipp}$N)$_2$U(NMeEt)$_2$ in a catalyst-to-monomer ratio of 1:60,000 was not inhibited, presumably due to the large cone angle of the ligands in the complex, impeding chain coordination and enabling only polymerization growth by inserting small monomers [26]. Hence, we decided to continue our studies using complex **4**.

The polymerization results of complex **4** are summarized in Table 3. Analysis of the polymers shows that even at very high cat:monomer ratios, the molecular weight of the polymer ($M_n$) increases, indicating that a single site complex is responsible for the reactivity [120].

**Table 2.** Polymerization results for the ROP of $\varepsilon$-caprolactone mediated by complexes **1–4**.

| Complex | Activity (g mol$^{-1}$ h$^{-1}$)·10$^4$ | | |
|---|---|---|---|
| Cat:Monomer | 1:100 | 1:1000 | 1:2000 |
| 1 | 4.5 | 11.4 | 6.1 |
| 2 | 3.9 | 9.8 | 6.3 |
| 3 | 2.5 | 4.9 | 4.4 |
| 4 | 10.7 | 19.8 | 27.5 |

Polymerization conditions: 41.67 µL of toluene, r.t, and 3 µmol of catalyst. The conversion was determined by $^1$H NMR spectroscopy of the crude reaction. Cat: catalyst.

$\varepsilon$-Caprolactone polymerized using complex **4** with a catalyst:monomer ratio of 1:100 in a toluene solution (5 mL) to avoid the transesterification that occurs with high monomer concentrations. Plotting of the polymerization conversion values obtained against the measured $M_n$ suggests that polymerization occurs in a linear fashion (R = 0.969), indicating that the monomer continues to be consumed as the polymerization propagates (Chart 1A). However, plotting the $M_n$ against time (Chart 1B) indicates that this process is not a living polymerization (R = 0.78), with PDI values ranging between 1.3 and 1.9 (Chart 1A). This result suggests a single-site catalyst with almost no transesterification side reactions.

**Table 3.** Polymerization results for the ROP of $\varepsilon$-caprolactone mediated by complex **4**.

| Cat:Monomer | 1:100 | 1:1000 | 1:2000 |
|---|---|---|---|
| Yield ($\pm$0.03%) | 78.3 | 14.6 | 10.8 |
| A (g mol$^{-1}$ h$^{-1}$)·10$^5$ | 1.07 | 1.98 | 2.75 |
| PDI | 1.7 | 2.4 | 2.7 |
| $M_{n\ corrected}$·10$^3$ (g/mol) | 10.9 | 13.0 | 15.8 |

Polymerization conditions: 41.67 µL of toluene, r.t, 3 µmol of catalyst, 5 min. The conversion was determined by $^1$H NMR spectroscopy of the crude reaction. The $M_n$ values were relatively calibrated by GPC using polystyrene standards; the $M_n$ values were multiplied by a factor of 0.58 and correlated to the actual PCL values [26,121,122].

The $^1$H NMR of the stoichiometric reactions between complex **4** and 1 equiv of $\varepsilon$-caprolactone (Figure S17) or 2 equiv of $\varepsilon$-caprolactone (Figure S18) revealed the existence of a caprolactonyl moiety at 5.82 ppm as the end group of the polymer, in addition to the appearance of the one or two equiv of C$H_3$-phenyl (toluene) as benzyl is the activated leaving group. In addition, we performed poisoning experiments by adding 1 equiv and 2 equiv of deoxygenated water to complex **4**, followed by the addition of monomer.

**A**

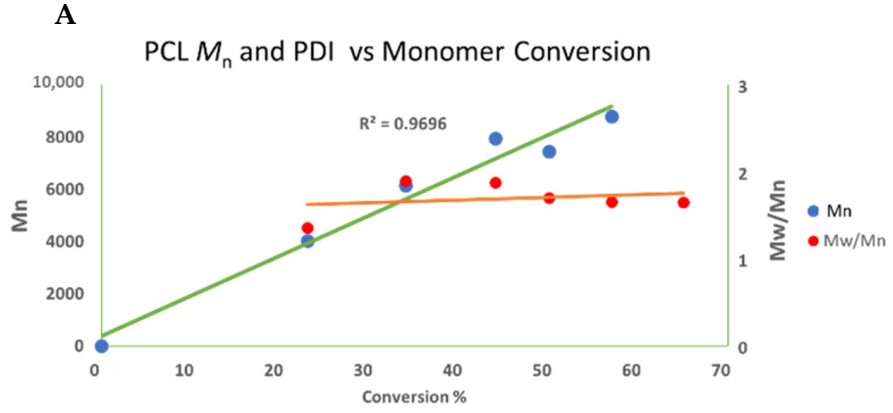

**Chart 1.** *Cont.*

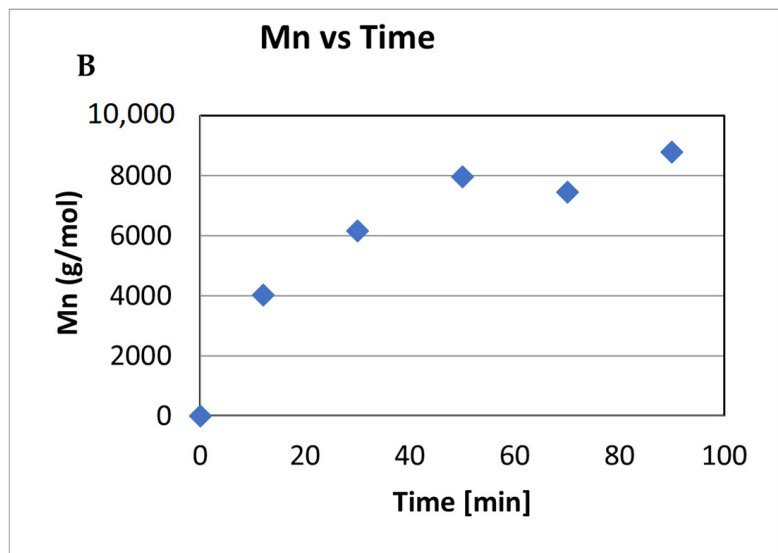

**Chart 1.** $M_n$ and PDI in the polymerization of caprolactone with complex **4** (**A**). $M_n$ as a function of time (**B**): 5 mL of toluene, r.t, 11 μmol of catalyst **4**. Cat:ε-caprolactone; 1:100. The yield was determined by [1]H NMR spectroscopy of the crude reaction. The Mn values were relatively calibrated using GPC with polystyrene standards; the $M_n$ values were multiplied by a factor of 0.58 and correlated to the actual PCL values [26,121,122].

In the case of 1 equiv of water, the polymerization proceeds as previously, showing a similar conversion as compared to the results obtained without water, suggesting that not all the benzyl groups are active in the polymerization. Performing the polymerization after adding 2 equiv of water inhibits the polymerization. This result strongly indicates that one active site is needed for polymerization to proceed.

Because in the polymerization of ε-caprolactone performed by complex **4**, the Hf-Bn motif is transformed into an Hf-OR group, we decided to study complexes **5** and **6**. Polymerization with complex **7** is almost identical to that with complex **5,** so we did not investigate this further. We decided to start with an oxophilic complex with a very strong Hf-OR bond because polymerization was started with am Hf-O motif that propagates through the polymerization, resulting in a similar Hf-O bond. Therefore, there is not an energetic cost for breaking such a strong bond. Hence, the polymerization carried out with complex **5**, (Chart 2) describes the conversion versus the molecular weight ($M_n$), showing a linear trend with a very narrow PDI, indicating that the polymerization proceeds in a living fashion, as demonstrated by the obtained $M_n$ of the polymer after 100% conversion. In addition, this result indicates that all the Hf-O motifs are active during polymerization.

[1]H NMR analysis of the polymer shows the $CH_2$ moiety belonging to the benzyl alcohol end group (Figure S21) at 5.11 ppm. Moreover, the aromatic hydrogens are found at 7.35 ppm, correlating well with the integration values (5:2). An additional chain-end group, a hydroxyl, was observed at 3.64 ppm (Figure S20) after the reaction mixture was washed with methanol.

Additional studies were performed using complex **6**. The obtained results were similar to those obtained with complex **5**, also displaying a linear trend in the plot of conversion versus $M_n$ (R = 0.986) (Chart 3), with very narrow polydispersity. Based on the chart values, a living polymerization of ε-caprolactone is obtained, catalyzed by complex **6**. These results indicate the high probability that complexes **5** and **6** are similar in terms of structure/active sites. Examination of the [1]H NMR spectrum demonstrated the presence of the distinctive end-group $(CH_3)_2CH\text{-}O\text{-}$ of the $^i$PrO group at 4.88 ppm ("g" in Figure S22). The additional end group, i.e., the hydroxyl group, was conspicuously located at 3.41 ppm, which was confirmed using $D_2O$ [123]. It seems that the mechanism of the polymerization for complexes **5**–**8** is the same as those reported in the literature with alkoxo complexes.

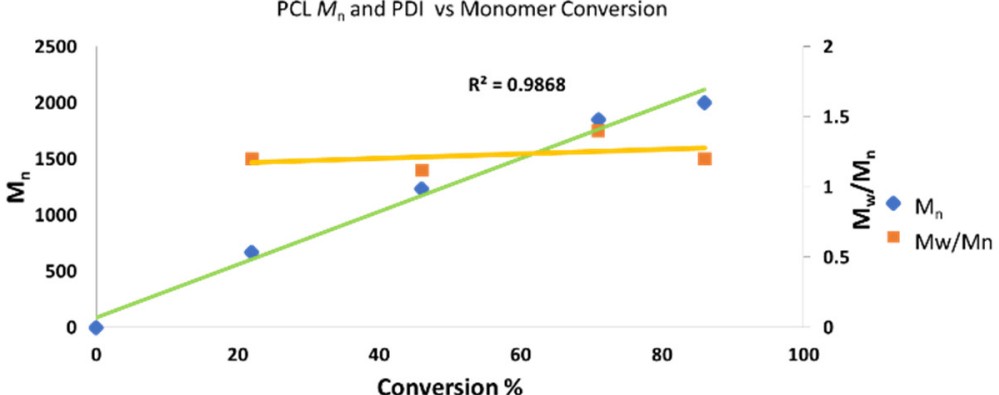

**Chart 2.** Linear increase in the molecular weight Mn as a function of conversion and polydispersity in the polymerization of ε-caprolactone catalyzed by complex **5**. Conditions: 5 mL of toluene, r.t, 11 μmol of catalyst 5. Cat: ε-caprolactone; 1:100 The yield was determined by [1]H NMR spectroscopy of the crude reaction. The $M_n$ values were relatively calibrated using a GPC with polystyrene standards; the $M_n$ values were multiplied by a factor of 0.58 [26,121,122].

**Chart 3.** Linear increase in the molecular weight $M_n$ as a function of conversion and polydispersity in the polymerization of ε-caprolactone catalyzed by complex **6**. Conditions: 5 mL of toluene, r.t, 11 μmol of catalyst **6**. Cat: ε-caprolactone; 1:100. The yield was determined by [1]H NMR spectroscopy of the crude reaction. The $M_n$ values were relatively calibrated using a GPC with polystyrene standards; the $M_n$ values were multiplied by a factor of 0.58 [26,121,122].

Complexes **5** and **6** carry alkoxide moieties, as well as the symmetric imidazolin[Dipp]-2-iminato ligand. The exchange of the benzylic groups (complex **4**) for a benzyloxy group (complex **5**) afforded a higher TOF of $140 \times 10^5 \text{ h}^{-1}$, and a similar result was obtained when switching to the isopropoxide groups (complex **6**), resulting in a TOF of $130 \times 10^5 \text{ h}^{-1}$. As expected, complexes **5** and **6** provided polymers with low molecular weights as compared to the polymers obtained by complex **4** due to their living fashion divided by the number of active alkoxy units. This result is also the outcome of the different mechanisms for the polymerizations. Whereas polymerization with complexes **5** and **6** proceeds in a living fashion, protonolysis of the chain is observed in the case of complex **4**, producing a caprolactonyl end group. The polymerization data are summarized in Table 4, and a plausible mechanism is presented in Scheme 4.

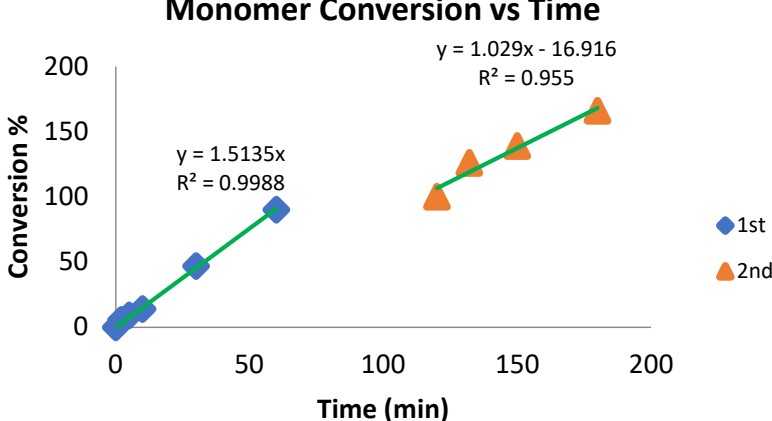

**Scheme 4.** A plausible mechanism for the polymerization of caprolactone with complex **4**.

The mechanism presented in Scheme 4 is based on J-Young NMR stoichiometric experiments (see supporting information). The first step is the protonolysis of the benzyl group with the $\alpha$-hydrogen of the caprolactone, as observed in the evolution of toluene in the NMR. The enolate hafnium is rapidly isomerized to the alkoxo complex, which is the active species. Additional insertions of caprolactone induce the formation of the polymeric chain, and cleavage via the same $\alpha$-hydrogen acidic protonolysis releases the polymer and regenerates the active species after the rapid rearrangement. The caprolactonyl terminal group is easy to observe (the double bond) in the NMR of the polymer (see SI, spectra S20).

A comparison of complexes **6** and **8** reveals the effect of the imidazolin-2-iminato ligand. Complex **6** was found to be more active than complex **8**, consuming 100% of the monomer, whereas **8** consumes only 58% of the monomer at the same time (after 30 min, complex **8** afforded a polymer with $M_n$ = 69,000).

Because complex **6** allows for living polymerization, we studied its performance upon a second addition of the monomer. We ran the polymerization reaction of $\varepsilon$-caprolactone with complex **6** (20:1 respectively), monitoring the reaction via $^1$H NMR. After all the monomer was polymerized, a second amount of $\varepsilon$-caprolactone was added, which was polymerized at a slightly lower rate, probably due to the increase in viscosity (Chart 4).

**Monomer Conversion vs Time**

y = 1.029x - 16.916
$R^2$ = 0.955

y = 1.5135x
$R^2$ = 0.9988

◆ 1st
▲ 2nd

**Chart 4.** Polymerization of $\varepsilon$-caprolactone displaying the conversion as a function of time. After reaching 100% yield, an equal amount of $\varepsilon$-caprolactone was added. Conditions: 5 mL of toluene, r.t, 35 μmol of catalyst **6**. Cat: $\varepsilon$-caprolactone; 1:20 for the first and second additions. The yield was determined by $^1$H NMR spectroscopy of the crude reaction.

**Table 4.** Polymerization results for the ROP of $\varepsilon$-caprolactone mediated by complexes **4–6**.

| Complex | Time (min) | Conversion % | $M_n$ (g/mol) | PDI |
|---------|-----------|--------------|---------------|-----|
| 4 | 10 | 23 | 4030 | 1.36 |
| 4 | 30 | 34 | 6160 | 1.90 |
| 4 | 50 | 44 | 7950 | 1.88 |
| 4 | 70 | 50 | 7450 | 1.70 |
| 4 | 90 | 57 | 8790 | 1.66 |
| 5 | 10 | 38 | 1120 | 1.25 |
| 5 | 20 | 60 | 1340 | 1.08 |
| 5 | 30 | 78 | 1680 | 1.10 |
| 5 | 50 | 95 | 1900 | 1.50 |
| 5 | 60 | 100 | 2130 | 1.10 |
| 6 | 10 | 22 | 670 | 1.20 |
| 6 | 20 | 46 | 1230 | 1.12 |
| 6 | 30 | 71 | 1850 | 1.40 |
| 6 | 40 | 86 | 2000 | 1.20 |

Polymerization conditions: 5 mL of toluene, r.t, 11 μmol of catalyst. Cat: $\varepsilon$-caprolactone; 1:100. The conversion was determined by $^1$H NMR spectroscopy of the crude reaction. The $M_n$ values were relatively calibrated using polystyrene standards; the $M_n$ values were multiplied by a factor of 0.58 (Mark–Houwink coefficient) and correlated to the actual PCL values.

We decided to study the ROP of *rac*-lactide to get a better understanding of whether the ability of complex **4** to polymerize would be compromised in a more crowded oxo environment, enabling the chelation of the monomer or the last inserted monomer unit to the metal center.

*rac*-Lactide polymerization with complex **4** indicates that the reaction proceeds in a linear fashion (Chart 5) according to the plot of $M_n$ versus the conversion to produce an atactic polymer, as expected. However, the rate of the reaction is slower as compared to PCL polymerization. The polymerization of the *rac*-lactide by complex **6** (Chart 6) is much faster, suggesting facile access of the monomer to the active site.

We were interested to explore the possibility of obtaining block copolymer PCL-*b*-PLA. This polymerization was performed by a method similar to that described for the immortal $\varepsilon$-caprolactone experiment. A Schlenk tube was charged with a solution of complex **4** dissolved in toluene, followed by the addition of 3 equiv of $^i$PrOH to produce the "in situ" complex **6**. Then, a measured amount of $\varepsilon$-caprolactone was added, and after complete consumption, the corresponding amount of *rac*-lactide was added. $\varepsilon$-caprolactone was polymerized at room temperature, whereas the lactide was polymerized at 70 °C (Chart 7).

**PLA Mn and PDI vs Monomer Conversion**

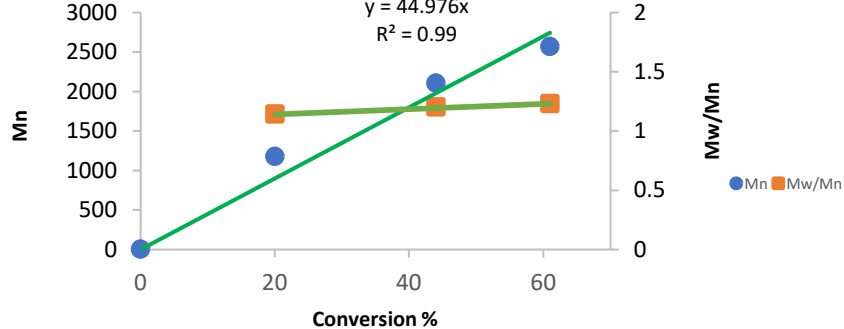

**Chart 5.** $M_n$ and PDI in the polymerization of *rac*-lactide with complex **4**. Conditions: 5 mL of toluene, 70 °C, 8.2 μmol of catalyst **4**. Cat: *rac*-lactide; 1:200. The conversion was determined by $^1$H NMR spectroscopy of the crude reaction. Mn values were relatively calibrated using GPC with polystyrene standards; the $M_n$ values were multiplied by a factor of 0.56 and correlated to the actual PLA values [124,125].

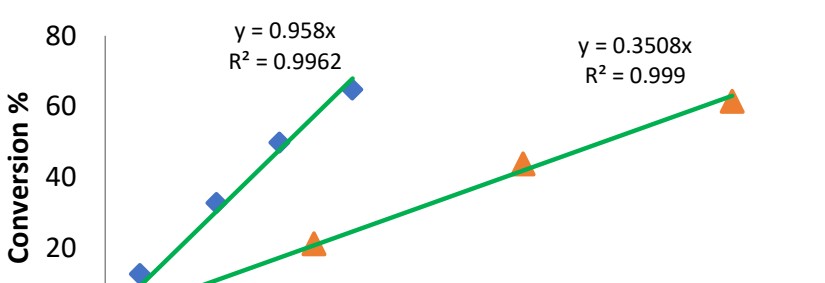

**Chart 6.** Conversion (%) in the polymerization of *rac*-lactide with complexes **4** (▲) and **6** (♦). 5 mL of toluene, 70 °C, 8.2 μmol of catalysts **4** and **6**. The conversion was determined by $^1$H NMR spectroscopy of the crude reaction. Ratio of catalyst **4**:*rac*-lactide, 1:200; ratio of catalyst **6**:*rac*-lactide, 1:300.

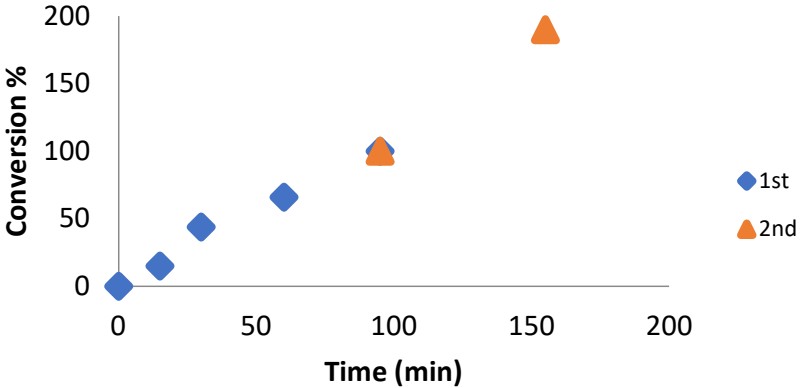

**Chart 7.** Polymerization conditions: 5 mL of toluene, 25 °C, 11 μmol of catalyst. Cat: ε-caprolactone; 1:100 for the first addition. The ratio of lactide monomer for the second addition was the same as in the first addition; 70 °C; the yield was determined by $^1$H NMR spectroscopy of the crude reaction mixture.

The $^1$H NMR spectrum of the formed copolymer was measured 55 min after the lactide addition, indicating that 90% of the lactide monomer was consumed (Chart 7). In addition, the spectrum indicates the appearance of both PCL and PLA in the copolymer. GPC analysis of each polymer/copolymer formed after each addition indicates that the polymer weight of the PCL was $M_n$ = 5000 Dalton, with a PDI of 1.1; after the lactide addition, the $M_n$ of the copolymer increased to 15,000, with a similar polydispersity of 1.2. (Figure S23). To ensure that only one type of polymer was obtained rather than mixtures of two homopolymers, PCL and PLA, a diffusion-ordered spectroscopy (DOSY) experiment was performed. This experiment allowed us to determine whether there was more than one polymer type, as polymers with different masses possess different diffusion coefficients. The DOSY experiment (Figure S24) revealed that there was only one copolymer type present.

### 3. Experimental Section

*3.1. General Considerations*

All manipulations of air-sensitive materials were performed with the rigorous exclusion of oxygen and moisture in flamed Schlenk-type glassware on a high vacuum line ($10^{-5}$ Torr) or in nitrogen-filled MBraun and vacuum atmosphere gloveboxes with a medium-capacity recirculator (1–2 ppm oxygen). Argon and nitrogen were purified by passage through am MnO oxygen-removal column and a Davison 4 Å molecular sieve column. Analytically pure solvents were dried and stored with Na/K alloy and degassed by three freeze−pump−thaw cycles prior to use (hexane, toluene, benzene-d6, and toluene-d8). Im$^{Dipp}$NH, Im$^{Mes}$NH, Im$^{tBu}$NH, Im$^{iPr}$NH, and the metal complex precursor hafnium tetrabenzyl were synthesized according to published literature procedures [125–127]. $\varepsilon$-CL was distilled and stored in the glovebox before use. *rac*-Lactide was recrystallized from Et$_2$O. $^{i}$PrOH, BnOH was distilled under CaH$_2$ and stored over 4Å molecular sieves in a glovebox prior to use. NMR spectra were recorded on Avance 200, Avance 300, Avance III 400, Avance 500, and Avance III 600 Bruker spectrometers (Karlsruhe, Germany). Chemical shifts for $^1$H NMR and $^{13}$C NMR measurements are reported in ppm and referenced using residual proton or carbon signals of the deuterated solvent relative to tetramethylsilane. MS experiments were performed at 200 °C (source temperature) on a Maxis Impact (Bruker) mass spectrometer with an APPI solid probe method. For X-ray crystallographic measurements, the single-crystalline material was immersed in perfluoropolyalkylether oil, quickly fished with a glass rod, and mounted on a Kappa CCD diffractometer under a cold stream of nitrogen. Data were collected using monochromated Mo K$\alpha$ radiation with $\varphi$ and $\omega$ scans to cover the Ewald sphere. The structure was solved by SHELXS-97 direct methods and refined by the SHELXL-97 program package 128-130]. The deposition CCDC number of complexes **1**, **3**, **5**, **7**, and **8** are 1,945,369, 1,945,366, 1,946,808, 1,945,247, and 1,946,807, respectively. These data are provided free of charge by the joint Cambridge Crystallographic Data Centre and Fach Information Zentrum Karlsruhe Access Structures Service www.ccdc.cam.ac.uk/structures (accessed on 1 September 2022).

GPC measurements were conducted on a Waters Breeze system with a styrogel RT column with THF (HPLC grade, T.G. Baker (Vaughan, ON, Canada)) as a mobile phase at room temperature. Relative calibration was performed with polystyrene standards (Aldrich, 2000–1,800,000 Dalton). Mn values were multiplied by a factor of 0.56 and correlated to actual PCL values [26,122]. Additional measurements for gel permeation chromatography (GPC) analyses were performed using tetrahydrofuran (THF) as a solvent at room temperature, using a Thermo LC system equipped with one Tosoh's TSKgel guard column HHR-L and four TSKgel G4000HHR columns in sequence. Detection was performed with a penta-detector system, including a Dionex DAD-3000 PDI UV–Vis detector, a Wyatt Viscostar II, Wyatt OPTILAB T-rEX, Wyatt MALS DAWN HELEOS II 8 + TR, and Wyatt QELS DLS. Wyatt's Astra 7.1.4 software was used for GPC data analysis and to calculate polymer properties (molecular weights, PDIs, intrinsic viscosities, and hydrodynamic radii). Molecular weights were calculated using Zimm plots with data from MALS and RI detectors.

*3.2. Synthesis of (Imidazolin-2-iminato) Hafnium (IV) Complexes **1–4**:*

Synthesis was performed following the previously published procedure for complexes **2** and **4** [68]. A toluene solution of the respective (imidazolin-2-imine) Im$^R$NH (0.184 mmol) in 5 mL of toluene was added dropwise to a pre-prepared solution of hafnium tetrabenzyl (100 mg 0.184 mmol) in toluene (5 mL) at room temperature. The reaction mixture was stirred overnight at room temperature. The solvent was removed under vacuum to afford crude complexes **1–4**. In each case, the product was recrystallized from a concentrated toluene solution at −35 °C to yield complexes **1–4** as crystalline materials.

[(Im$^{iPr}$N)Hf(Bn)$_3$], (Complex **1**): Yield: 98 mg, 0.158 mmol, 86%.

$^1$H NMR (300 MHz, C$_6$D$_6$) δ 7.06 (m, 15H, *H*-Ar), 5.78 (s, 2H, NC*H*), 4.14 (m, 2H, C*H*(CH$_3$)$_2$), 2.08 (toluene), 1.20 (s, 6H, Hf-C*H*$_2$-Ph), 0.94–0.86 (m, 12H, C*H*$_3$). $^{13}$C NMR

(75 MHz, $C_6D_6$) δ 152.64, 141.42, 137.39, 128.83, 125.72, 125.20, 106.17, 43.99, 37.71, 20.90. Elemental analysis calculated for $C_{60}H_{74}Hf_2N_6$: C, 58.29; H, 6.03; N, 6.80; found: C, 58.67; H, 5.74; N, 6.99. MS(APPI) calculated for $C_{60}H_{74}Hf_2N_6$ (M + $H_2O$ from the APPI)) = 1253.4978, Found = 1253, 4959.

[(Im$^{t}$$^{Bu}$N)Hf(Bn)$_3$], (Complex **2**): Yield: 97 mg, 0.151 mmol, 82%.

$^1$H NMR (300.0 MHz, $C_6D_6$) δ 7.19 (t, $^3$J = 7.7 Hz, 6H, *m*-Ar), 6.97 (t, $^3$J = 7.4 Hz, 3H, *p*-Ar), 6.88 (d, $^2$J = 7.3 Hz, 6H, *o*-Ar), 5.91 (s, 2H, NC*H*), 1.94 (s, 6H, Hf-C*H*$_2$-Ph), 1.39 (s, 18H, CC*H*$_3$). $^{13}$C NMR (150.0 MHz, $C_6D_6$) δ 143.75, 137.1, 128.98, 127.79, 122.15, 107.13, 75.05, 55.84, 27.79. Elemental analysis calculated for $C_{32}H_{41}HfN_3$: C, 59.48; H, 6.40; N, 6.50; found: C, 59.33; H, 5.51; N, 6.76.

[(Im$^{Mes}$N)Hf(Bn)$_3$], (Complex **3**): Yield: 127 mg, 0.165 mmol, 90%.

$^1$H NMR (600.0 MHz, $C_6D_6$) δ 7.52 (s, 4H, *m*-Ph), 7.43 (t, $^3$J = 7.7 Hz, 6H, *m*-Ph), 7.27 (t, $^3$J = 7.4 Hz, 3H, *p*-Ph), 6.82 (d, $^2$J = 7.1 Hz, 6H, *o*-Ph), 6.01 (s, 2H, NC*H*), 2.52 (s, 12H, *o*-C*H*$_3$Ph), 2.46 (s, 6H, *p*-C*H*$_3$Ph), 1.62 (s, 6H, Hf-C*H*$_2$-Ph). $^{13}$C NMR (150. 0 MHz, $C_6D_6$) δ 143.81, 142.97, 138.69, 136.50, 133.12, 129.19, 129.02, 127.36, 121.90, 112.09, 69.08, 20.65, 17.67. Elemental analysis calculated for $C_{42}H_{45}HfN_3$: C, 65.49; H, 5.89; N, 5.45; found: C, 64.93; H, 5.40; N, 5.89. MS(APPI) calculated for $C_{42}H_{45}HfN_3$ (M+ Na from the APPI)) = 794.2977, found = 794.2782.

[(Im$^{Dipp}$N)Hf(Bn)$_3$], (Complex **4**): Yield: 149 mg, 0.175 mmol, 95%.

$^1$H NMR (300.0 MHz, $C_6D_6$) δ 7.10–7.15 (m, 6H, Ar*H*), 7.03 (t, $^3$J = 7.5 Hz, 6H, *m*-Ph), 6.85 (t, $^3$J = 7.3 Hz, 3H, *p*-Ph), 6.35 (d, $^2$J = 7.0 Hz, 6H, *o*-Ph), 5.91 (s, 2H, NC*H*), 4.33 (s, 4H, PhC*H*(CH$_3$)$_2$), 1.35 (d, $^2$J = 6.9 Hz, 12H, C*H*$_3$), 1.17 (s, 6H, HfC*H*$_2$Ph), 1.10 (d, $^3$J = 6.9 Hz, 12H, C*H*$_3$). $^{13}$C NMR (150.0 MHz, $C_6D_6$) δ 146.91, 142.61, 133.48, 129.73, 129.08, 127.66, 127.44, 123.94, 121.91, 113.78, 70.24, 28.67, 23.92, 23.20. Elemental analysis calculated for $C_{48}H_{57}HfN_3$: C, 67.47; H, 6.72; N, 4.92; found C, 67.52; H, 6.85; N, 4.94.

*3.3. Synthesis of (2,6-diisopropylphenyl)imidazolin-2-imine Hafnium tris BnO [(ImDippNH)Hf(BnO)$_3$], (Complex 5):*

A solution of BnOH (18 μL, 0.176 mmol) in 5 mL of toluene was added to a pre-prepared solution of complex **4** (50 mg, 0.058 mmol) in toluene (5 mL) at room temperature, and the reaction mixture was stirred overnight at room temperature. The solvent was removed under vacuum to afford crude complex **5**. The crude product of **5** was recrystal-lized from a concentrated toluene solution at −35 °C to yield 0.052 g, 0.058 mmol, and 99% crystalline material.

$^1$H NMR (600 MHz, $C_6D_6$) δ 7.19–6.96 (m, *H*-Ar), 5.72 (s, 2H, NC*H*), 4.34 (brH, 6H, OC*H*$_2$Ph), 3.03 (q, 4H, CC*H*(CH$_3$)$_2$), 1.11 (d, $^2$J = 6.9 Hz, 12H, C*H*$_3$), 1.01 (d, $^2$J = 6.9 Hz, 12H, C*H*$_3$). $^{13}$C NMR (150 MHz, $C_6D_6$) δ 147.40, 145.73, 137.55, 134.83, 129.19, 128.95, 128.17, 125.29, 124.00, 114.00, 31.59, 28.61, 25.43, 24.01, 23.69, 22.65, 21.04, 13.97. MS(APPI) for $C_{96}H_{114}Hf_2N_6O_6$ (M + 2H from the APPI)) = 1807.7892, found = 1807.7875. Elemental analysis calculated for $C_{96}H_{114}Hf_2N_6O_6$: C, 63.88; H, 6.37; N, 4.66; found C, 63.64; H, 6.58; N, 4.49.

*3.4. Synthesis of (2,6-diisopropylphenyl)imidazolin-2-imine Hafnium tris($^i$PrO) $^i$PrOH [(ImDippNH)Hf($^i$PrO)$_3$ ($^i$PrOH)], (Complex 6):*

A solution of $^i$PrOH (175 μL, 0.176 mmol) in 5 mL of toluene was added to a pre-prepared solution of complex **4** (50 mg, 0.058 mmol) in toluene (5 mL) at room temperature, and the reaction mixture was stirred overnight at room temperature. The solvent was removed under vacuum to afford crude complex **6**. The product was precipitated by the addition of 6 mL of hexane, and the solvent was removed by decantation to yield 0.039 g, 0.052 mmol, and 90% material. One molecule of the isopropanol was co-crystallized with the complex, and its proton was exchanged with the isopropoxide moieties. Hence, the isopropyl moieties were not equivalent, resulting in a broad spectrum. However, the MS and microanalysis corroborate the formation of the complex.

$^1$H NMR (300 MHz, C$_6$D$_6$) δ 7.20–6.32 (m, *H*-Ar), 5.80 (s, 2H, NC*H*), 4.68 (m, 3H, C*H*(CH$_3$)$_2$), 4.11 (s, 1H, C*H*(CH$_3$)$_2$), 3.12 (m, 4H, C*H*(CH$_3$)$_2$), 1.41 (d, $^2$J = 6.4 Hz, 18H, C*H$_3$*), 1.34(d, $^2$J = 6.2 Hz, 6H, C*H$_3$*), 1.16 (m, 24H, C*H$_3$*).

$^{13}$C NMR (75 MHz, C$_6$D$_6$) δ 154.08, 146.89, 142.61, 133.43, 129.11, 128.84, 125.18, 123.89, 121.90, 70.83, 36.60, 31.44, 28.54, 26.28, 23.46, 20.91, 13.86. HRMS(APPI) for C$_{39}$H$_{65}$HfN$_3$O$_4$ + H (from the APPI) (M + H) = 819.6044. Elemental analysis calculated for C$_{39}$H$_{65}$HfN$_3$O$_4$: C, 57.23; H, 8.01; N, 5.13; found C, 56.98;. H, 8.48; N, 4.99.

### 3.5. Synthesis of Tetrakis Benzyloxy Hafnium, [(Hf(OBn)$_8$(ROH)], (Complex **7**):

A solution of BnOH (48 μL, 0.460 mmol) in toluene (5 mL) was added to a prepared solution of the corresponding hafnium tetrabenzyl (50 mg, 0.092 mmol) in 5 mL of toluene at room temperature. The reaction mixture was stirred overnight at room temperature. The solvent was removed under vacuum to afford crude complex **7**. The crude product of **7** was recrystallized from a concentrated toluene solution at room temperature.

[Hf$_2$(OBn)$_8$(BnOH)], (Complex **7**): Yield: 53 mg, 0.081 mmol, 88%.

$^1$H NMR (500 MHz, C$_6$D$_6$) δ 7.12–7.04 (45H, m, *H*-Ar), 4.35 (s, 12H, OC*H$_2$*Ph), 4.28(s, 6H, OC*H$_2$*Ph). $^{13}$C NMR (126 MHz, C$_6$D$_6$) δ 141.10, 128.06, 127.74, 127.54, 127.35, 126.95, 126.70, 64.17. MS(APPI) for C$_{63}$H$_{64}$Hf$_2$O$_9$ (M-HOBn) = 1321.3448 found = 1321.34854 (low intensity). Elemental analysis calculated for C$_{63}$H$_{64}$Hf$_2$O$_9$: C, 57.23; H, 4.88; found C, 56.94; H, 5.02.

### 3.6. Synthesis of Tetrakis Isopropoxy Hafnium, [(Hf(O$^i$Pr)$_8$($^i$PrOH)$_2$], (Complex **8**):

Hafnium tetrabenzyl (50 mg, 0.092 mmol) was dissolved in 5 mL of toluene at room temperature. In addition, a solution of iPrOH was prepared by the addition of iPrOH (32 μL, 0.460 mmol) to 5 mL of toluene. The iPrOH solution was added dropwise to the hafnium tetrabenzyl solution under vigorous stirring at room temperature. The reaction was stirred overnight before the solvent was removed under vacuum, resulting in crude complex **8**. The crude product of **8** was crystallized from a concentrated hexane solution at room temperature. [Hf$_2$(O$^i$Pr)$_8$($^i$PrOH)$_2$], (Complex **8**): Yield: 40 mg, 0.085 mmol, 93%. The rapid exchange of the proton hydrogens resulted in a very broad NMR. Cooling the solution precipitated the complex, and complex coalescence was obtained by heating. Additional heating decomposed the complex.

$^1$H NMR (500 MHz, C$_6$D$_6$) δ 4.64 (m, 1H, C*H*(CH$_3$)$_2$), 1.40 (m, 6H, C*H$_3$*). $^{13}$C NMR (126 MHz, C$_6$D$_6$) δ 69.31, 26.30. MS(APPI) calculated for C$_{30}$H$_{72}$Hf$_2$O$_{10}$ (M) = 950.4028, found = 950.4084.

Elemental analysis calculated for C$_{30}$H$_{72}$Hf$_2$O$_{10}$: C, 37.93; H, 7.64; found C, 38.04; H, 8.44.

### 3.7. ROP of ε-Caprolactone

A toluene solution of 3 mL charged with ε-caprolactone (130 μL, 1.1 mmol) was injected with a 2 mL toluene solution with complex **6** (9 mg,11 μmol). At the appropriate time, 0.4–0.5 mL samples were removed with a pipet to a vial and immediately removed from the glovebox and quenched with cold methanol. These samples were taken to the NMR to measure conversion via $^1$H NMR analysis. Afterward, these samples were dried under a vacuum and measured by GPC. The polymerization reaction was performed in a glovebox at room temperature.

### 3.8. ROP of rac-Lactide

A Schlenk tube charged with *rac*-lactide (170 mg, 1.1 mmol) and 3 mL of toluene was prepared in the glovebox and transferred to a preheated oil bath at 70 °C until all the lactide was dissolved. Then, the lactide solution was injected into a 2 mL toluene-containing complex (**6**) (9 mg, 11 μmol) under a nitrogen stream. The reaction was monitored by samples taken from the reaction. The molecular weight was measured by GPC.

*3.9. PCL-b-PLA*

The copolymerization reaction was started with a solution of 3 mL of ε-CL (77 μL, 0.7 mmol) dissolved in toluene. The monomer solution was added to a vial charged with complex **6** (27 mg, 35 μmol) and 2 mL of toluene. The characterization procedure was performed as mentioned previously. After monomer consumption was achieved, the mixture was heated to 70 °C. A Schlenk tube with *rac*-lactide (100 mg, 0.7 mmol) was charged in the glovebox and transferred and heated in an oil bath to 70 °C until the lactide was dissolved. Afterward, the lactide solution was injected into the first polymerization reaction. The reaction was monitored by samples taken from the reaction and quenched with methanol out of the glovebox. The molecular weight was measured by GPC.

**4. Conclusions**

In this work, we described the structural properties of several imidazolin-2-iminato hafnium complexes. These complexes differ in terms of their steric properties, which influence their reactivity towards the ROP of cyclic esters. All investigated catalysts showed tolerance toward oxygen-containing monomers, such as ε-caprolactone and *rac*-lactide. Owing to their stability, the complexes were also found to be suitable for block copolymerization between ε-caprolactone and *rac*-lactide. Therefore, it is possible to tailor the oxophilicity of hafnium complexes, allowing the new complexes to be active catalysts with oxygen-containing substrates.

**Supplementary Materials:** The following supporting information can be downloaded at: https://www.mdpi.com/article/10.3390/catal12101201/s1. The supplementary material contains: 1. General Considerations, 2. General Procedure for the Synthesis of Mono(imidazolin-2-minato) Hafnium(IV) Complexes, 3. [1]H NMR and [13]C NMR of Complexes. 3.1. Complex **1**, 3.2. Complex **2**, 3.3. Complex **3**, 3.4. Complex **4**, 3.5. Complex **5**: 3.6. Complex **6**, 3.7. Complex **7**, 3.8. Complex **8**. 4. [1]H NMR of Polymers, 4.1. PCL, 4.1.1. Stochiometric Reaction—Complex **4**: ε-Caprolactone; 1:1, 4.1.2. Stochiometric Reaction—Complex **4**: ε-Caprolactone; 1:2, 4.1.3. [1]H NMR of PCL Performed by Complex **4**: ε-Caprolactone, 1:100, 4.1.4. [1]H NMR of PCL Performed by Complex **5**: ε-Caprolactone, 1:100—End Group. 4.1.5. [1]H NMR of PCL Performed by Complex **6**: ε-Caprolactone, 1:100—End Group, 4.2. PLA, 4.2.1. [1]H NMR of PLA Performed by Complex **6**: *rac*-lactide, 1:100, 4.3. PCL-b-PLA, 4.3.1. 2D-DOSY NMR of the PCL-b-PLA, 4.4. D$_2$O Experiments, 4.5. Kinetic Graph and MALDI-TOF Spectra, 5. Crystallographic Data of complexes **1**, **3**, **5**, **7**, **8**, 6. References [26,68,107,121,126–130] are cited in the Supplementary Materials.

**Author Contributions:** Conceptualization, M.S.E., H.L. and M.T. Funding Acquisition, M.S.E.; Investigation, M.K.; X-ray Analysis, N.F.; writing—original draft preparation, M.S.E., H.L. and M.T.; Writing, M.S.E.; Administration, M.S.E. All authors have read and agreed to the published version of the manuscript.

**Funding:** This work was supported by the Israel Science Foundation, administered by the Israel Academy of Science and Humanities under Contract Number No. 184/18, and by the PAZY Foundation Fund under Contract Number 128-19 (2020) administered by the Israel Atomic Energy Commission.

**Conflicts of Interest:** The authors declare no conflict of interest.

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
