# Peer review of "Mono(imidazolin-2-iminato) Hafnium Complexes: Synthesis and Application in the Ring-Opening Polymerization of ε-Caprolactone and rac-Lactide"

_catalysts, doi:10.3390/catal12101201_

Round 1
Reviewer 1 Report
The article “Mono(imidazolin-2-iminato) Hafnium Complexes: Synthesis and Application in the Ring-Opening Polymerization of É›-Caprolactone and rac-Lactide” describes the preparation of hafnium complexes using various imidazolin-2-iminato ligands. The activities of these complexes were investigated by ring-opening (co)polymerization of lactones such as É›-caprolactone and lactide The results are of common interest for scientists studying catalytic systems for the polymerization of cyclic esters. The manuscript needs a major revision before being considered for publication.
1. The authors declared the ROP of ε-caprolactone using complexes 1–4 is not living polymerization with almost no transesterification side reactions while those using complexes 5 and 6 proceeds via a living mechanism, however, most data were collected at monomer conversion values lower than 90% whereas transesterification usually occurs at a very low concentration of monomer (monomer conversion higher than 90%). The author should also implement the semi-logarithmic plot of monomer concentration vs polymerization time to determine the reaction order.
2. The theory Mn calculated by NMR should be included to further compare with those determined by GPC.
3. The MWs of the resultant polycaprolactones and polylactides are surprisingly low (Chart 2 and 3, Table 4 complexes 5 and 6), considering the monomer feed ratio and monomer conversion, please explain.
4. MALDI-TOF data of the polymers should be added to corroborate the mechanistic pathway.
5. The authors mentioned about complex 8 in lines 365–368, however, only polymerization results acquired by complexes 1–6 are given whereas those obtained by complex 7 and 8 are missing. I recommend the authors to implement 1H NMR spectra of all polymers in the Supporting Information to facilitate follow-up of comments made by the authors.
6. Did the authors try ROP of lactide at room temperature since the hafnium complexes exhibit good activity for the ROP of É›-caprolactone at room temperature.
Minor remarks:
-The proposed mechanism in Scheme 4 seems to be acquired by complex 5 or 7 which is prepared by adding BnOH. Please clarify this in the figure caption.
-Please replace uL and umol to μL and μmol, respectively in the Experimental Section.
-Please improve the quality of Figures S19–S21 in the Supporting Information.
Author Response
Please see attached reply document

Reviewer 2 Report
The manuscript should be considered for publication after revision.
Specific comments:
- The use of the numbers for referring to the different catalyst structures should be avoided in the abstract
- Most of the introduction section is focused on the discussion of the use of polyesters and their advantages as degradable and sustainable materials. However, this contribution is dedicated to the characterization and study of the catalytic activity of hafnium-based catalyst in ROP. I would suggest to shift the focus on the introduction on the relevance of this class of catalyst in ROP and their differences/comparisons with metallic systems already explored
- The size and quality of Scheme 1, Scheme 2, Figure 1, Figure 2, Figure 3, Figure 4, Scheme 3 should be enhanced to enable readability and understandability
- A greater effort should be dedicated to correlate the structural features of the catalysts to the differences observed for their activity in the ROP
- Page 9, lines 277-279. The authors claim that by increasing the ratio of monomer to initiator (data in Table 4) the molar mass of the polymers increases. Yet, the increase of molar mass is not so significant and not linearly with increasing the feed. The discussion should be amended and an explanation for that should be provided.
Author Response
Please see attached reply document

Reviewer 3 Report
This work by Liu, Eisen and co-workers describes the synthesis of Hf complexes bearing imidazolin-2-iminato ligands and their activity as catalysts in the ring opening (co)polymerization of ε-caprolactone and lactide. Structure/activity trends, as well as the properties of the resulting polymers have been discussed.
The development of efficient catalysts to produce biodegradable polymers has been gaining increasing interests and, despite the tremendous number of (outstanding) systems reported until now, I believe the topic is far from being completely explored. To the best of my knowledge, the authors report the first example of imidazolin-2-iminato-Hf complexes used as ROP catalysts. Given the performances of these species, as well as their rarity, I strongly believe the manuscript would be of interest to the readers of Catalysts, hence I recommend its publication.
Some (minor) issues to be assessed:
In the Abstract, the iPr group is not mentioned in the list of the possible substituents on the ligand framework: “(X= tBu, Mesityl, Dipp)”. It should also be stated that the metal center is bound to Bn- ancillary ligands in complexes 1-4. This would clarify which benzyl groups are replaced by adding an alcohol (lines 17 and 18 of the abstract).
The introduction and the part of the Results and Discussion regarding the synthesis and characterization of the complexes are clear and nicely written. I would suggest adding the following references: Catalysts, 2021, 11, 551; Nature 2016, 540, 354-362; Dalton Transactions, 2020, 49, 11978–11996.
In Scheme 3, the bridging BnO-groups in complex 5 appear as protonated. Since this is not mentioned in the text, I suppose there is an issue with the drawing.
In the caption to the same scheme, please add “spectroscopy” after “NMR”. This should be done everywhere throughout the manuscript.
The ROP section needs some improvement.
At the end of page 8, the authors claim that transesterification occurs with complexes 1-3 for higher monomer:catalyst ratios. How was this concluded, since no GPC data is shown for this set of reaction? It would be beneficial for the readers having a complete table indicating these values.
Please change “gr” to “g” to indicate “grams”.
Table captions indicate that NMR spectroscopy was employed to determine the polymerization yields. Yet, the supporting information state that the samples were isolated and dried after the analysis. Please clarify this point. If no internal standard was used in the NMR spectroscopy analyses, “conversion” would be more appropriate than “yield”.
Table 3 is rather difficult to read. I would suggest rearranging it (in particular removing the first line repeating the units of the activity). Also, indicate the units of Mn.
Page 9, lines 288-289: the Mn vs Time plot is not shown.
Concerning the poisoning tests, I wonder whether the water employed in the experiments was deoxygenated. Could oxygen be responsible for catalyst deactivation? If the addition of water generated an Hf-OH species analogue to Hf-OR, could it still be capable of catalyzing the reaction? Indeed, many ROP catalysts proved to be activated by adventitious water present in the system.
Caption to charts 2 and 3 indicate Mn vs time, while the x axis reports the monomer conversion.
Could the authors spend some words on the mechanism proposed in Scheme 4? Has the caprolactonyl end group been previously observed? Is there any other evidence supporting the mechanism, especially with respect to the C-H activation-like step generating toluene? In this respect, “[Hf]-OBn” group should be changed into “[Hf]-Bn”. More details should also be provided into the ESI regarding the stochiometric experiments. I assume these were carried out on a small scale into Young-valved NMR spectroscopy tubes, but this is not clarified anywhere. In addition, I suggest assigning signals directly on the spectra in Figures S17 and S18, as it has been done for the polymer spectra (i.e. figure S20)
Please uniform the size and format of all charts reported in the manuscript.
Lastly, the molecular weight distribution could also be indicated by the symbol “Д.
Author Response
Please see attached reply document

Round 2
Reviewer 1 Report
The authors have revised their manuscript by reflecting my comments. I think the manuscript is now improved enough to be published.
Reviewer 2 Report
The manuscript has been improved.